# LATENT LIE GROUP REPRESENTATIONS

## ABSTRACT

Symmetry detection tasks rely on identifying transformations of data points that keep some task-related quality, such as classification label, identical. These symmetries are useful during model selection for neural networks, as even a conceptually simple symmetry (e.g., translation invariance) can lead to superior performance-efficiency tradeoffs (e.g., CNNs). Leveraging neural networks to learn these transformations can lead to approaches that yield representations of the transformations in latent space, rather than just the data itself. In this work, we propose a latent variable framework for learning one-parameter subgroups of Lie group symmetries from observations, improving the accuracy of the learned transformation with respect to the one in pixel-space, even including situations in which this might not even be desirable.

## 1 INTRODUCTION

It is well-known that restricting the hypothesis space of a neural network, by exploiting known properties of the task, improves performance in a variety of machine learning settings (Cohen & Welling, 2016). The community has dedicated a lot of effort investigating the relationship between symmetries of a specific task and optimal model architecture, providing practical constraints of the hypothesis space while retaining the universal approximator property of deep learning (Kondor & Trivedi, 2018; Yarotsky, 2022). In computer vision, the challenge of designing novel neural network architectures are often related to ensuring the constrained model is invariant with respect to transformations that preserve the object label, such as translation, rotation, and scaling, but also object permutations. Many of these transformations are smooth and differentiable, and thus belong to the family of Lie groups, which is the class of symmetries we deal with in this work.

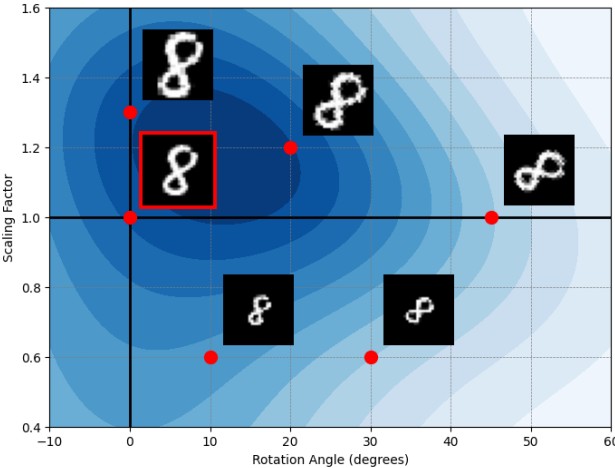

Figure 1: The distribution of data in a dataset that correspond to rotation and scaling, given in terms of the parameters degrees and scaling factor, respectively. These groups are differentiable and can be decomposed into one-parameter subgroups. (Taken from Gabel et al. (2023) with kind permission from the authors.)

## 2 BACKGROUND

First, we describe the problem setting and define a new task for symmetry detection. The task is contingent on the transformation and parameter distribution and therefore does not have a specific associated dataset (like Rotated MNIST) besides, of course, the source data. Then, our approach to solving the problem of symmetry detection using the formalism of Lie theory is explained, following an approach by Oliveri (2010) and recently applied to the same problem in Gabel et al. (2023). A more thorough introduction to the topic can be found in Fulton & Harris (1991) and Olver (1993), with a focus on representation theory and differential equations, respectively.

### 2.1 DEFINING THE TASK: SYMNIST

A symmetry is a transformation that leaves a certain quantity of interest unchanged. In order to define the symmetry under consideration, we must state what is being kept "the same". For the experiments that follow, we consider the classification task and the symmetry that keeps the underlying label identical. The general problem, therefore, is learning transformations that map instances of the same class (i.e., data points with the same underlying label) to each other. One can ask whether *global symmetries*, transformations that map data points to other "plausible" data points in the overall data distribution (for generative modelling purposes, perhaps) or *class-specific symmetries* are of interest as well[1]. This is indeed the case, but we will restrict to transformations which are symmetries within each class, but also common among all classes. One could call these *class-generic symmetries* and are a subset of the set of global symmetries.

For the experiments, we introduce the SyMNIST task. The dataset consists of MNIST data, paired with an augmented version of the original image. The original digit labels are not used in the current setting. The goal is to somehow extract the type of transformation that was applied, and the distribution in the form of what we call the "parameters" of the transformation (e.g., rotation angle, shift distance, scaling factor, etc.). As a transformation can be defined by a set of functions, it does not suffice to employ a table of predetermined symmetries (Benton et al., 2020), more flexibility is required. We wish to learn *non-canonical symmetries*, symmetries that are not restricted to vocabulary such as "rotation" or "translation", as well. Additionally, it is worth emphasizing that in a real setting, two images from the same class will necessarily not be related to one another in such a way, since in the SyMNIST task, the two images originate from the same underlying source image. In our experiments, the first in the pair of images is the source image itself, but one can easily generalize this by transforming the source image twice and learning the transformation that maps them to each other.

### 2.2 LIE GROUPS AND LIE ALGEBRAS

The transformations that describe the symmetries are assumed to form *Lie groups*. This means they are sufficiently smooth ($k$-times differentiable, where $k$ is usually chosen to be infinity), closed under composition, associative, have a neutral element, and have smooth inverse. These transformations can be defined by the way in which they *act* on objects, namely $\mathbf{H} : X \times \mathbb{R} \to X$, with $X \subset \mathbb{R}^n$. This also introduces a parameter which is associated to the magnitude of the transformation. For rotations, this will correspond to the angle, for translations, it will be the distance, etc. Because of continuity in the parameter, which we call $t$, we can expand for small values:

$$\mathbf{H}(\mathbf{x}, t) \approx \mathbf{x} + t\boldsymbol{\Gamma}(\mathbf{x}), \quad \boldsymbol{\Gamma}(\mathbf{x}) := \left.\frac{\partial \mathbf{H}(\mathbf{x}, t)}{\partial t}\right|_{t=0}. \tag{1}$$

Since the transformation forms a Lie group, we can use the First Fundamental Theorem of Lie (Olver, 1993; Oliveri, 2010) in order to make the following claim: $\boldsymbol{\Gamma}(\mathbf{x})$ defines the transformation and is related to the generator of the transformation. Intuitively, this correspondence between action and generator is due to the constraints imposed on the transformation function being a Lie group.

---

[1] For example, generating a new digit from MNIST data, versus flipping the digit "8" horizontally and vertically which yields another potential "8".

The generator can be written as a differential operator as follows:

$$G = \sum_{i=1}^{n} \Gamma_i(\mathbf{x}) \frac{\partial}{\partial x_i} \tag{2}$$

If one solves the differential equations that characterize the generator, the original transformation function is obtained. More specifically, the solution is the family of functions connected to the identity transformation. The generator is an element of the *Lie algebra* of the transformation group and is related to the original transformation by what is called the *exponential map*. This nomenclature emphasizes the connection between the differentiation performed in Equation 1 and exponentiation, easily seen when solving the characteristic equation (Olver, 1993), i.e.,

$$\frac{d\mathbf{H}(\mathbf{x}, t)}{dt} = G\mathbf{H}(\mathbf{x}, t), \quad \mathbf{H}(\mathbf{x}, 0) = \mathbf{x}, \tag{3}$$

as the solution is $\mathbf{H}(\mathbf{x}, t) = e^{tG}x$ with $e^{tG} := \sum_{k=0}^{\infty} \frac{1}{k!} t^k G^k$, where the integer power of $G$ is defined by applying it iteratively. The inverse procedure, which extracts the generator from the action as shown in Equation 1, is also referred to as the *logarithmic map*.

We focus on one-parameter groups for two reasons: Simplicity and the fact that one such inductive bias is incredibly powerful already. One need not look further than CNNs to see that identifying translation as a symmetry of a dataset immediately leads to equivariant models that are superbly successful in practice. Multiple transformations also require additional considerations that relate to the algebra itself, such as closure under commutators (Roman et al., 2023).

Learning connectivity matrices for deep equivariant models, whether the symmetry group is given or not, is not new (Finzi et al., 2021; Kondor & Trivedi, 2018; Zhou et al., 2020). It is worth noting that generators can be related to the connectivity matrix, explicitly so for translations, where the shift matrix determines a power series that tiles the weight matrix accordingly. Formally, for the one-pixel shift matrix $S = e^{\partial_x}$, we can write:

$$f_\theta^L = \sum_{i \in \mathbb{Z}} \theta_i S^i. \tag{4}$$

This defines one such convolutional layer. A collection of multiple power series applied in succession *is* the neural network. One wonders how this extends to other networks.

In this work, in order to eschew the potential computational complexity of the permutation group, another goal is to avoid making calculations in pixel-space directly. On the other hand, we don't want to introduce spatial correlation as a constraint a priori, as this defeats the purpose of learning the transformations from scratch. Additionally, this strict assumption does not allow for the possibility of detecting the negative case, namely one in which there is no underlying symmetry and the most appropriate model in the hypothesis space is a fully-connected neural network. Here, latent transformations are learned, and the pixel-level transformation will need to be extracted by special methods.

### 2.3 RELATED WORK

**Equivariance**    A lot of work has been dedicated to designing neural networks that are equivariant with respect to a given transformation (Kondor & Trivedi, 2018; Finzi et al., 2021). Transformations of interest beyond translation are scaling (Worrall & Welling, 2019; Sosnovik et al., 2021), rotation on spheres (Cohen et al., 2018), local gauge transformations (Cohen et al., 2019) and the Euclidean group (Weiler & Cesa, 2019), as well as discrete transformations like permutations of sets (Zaheer et al., 2017; Zhang et al., 2022) and time-reversal (Valperga et al., 2022). Research in these areas shows improved performance on tasks that are symmetric w.r.t. the transformation under consideration, but nonetheless requires knowledge about the symmetries a priori.

**Symmetry Detection**    Early work on detecting symmetries from observations was performed by Rao & Ruderman (1998) and Miao & Rao (2007), who use methods to learn transformations for small parameter values. Sohl-Dickstein et al. (2010) propose a smoothing operation of the transformation space to overcome the issue of a highly non-convex reconstruction objective that includes an exponential map. These methods are close to ours in that we also make use of the exponential

map to obtain group elements from their Lie algebra, although their work being focused on video patches and using EM-algorithms to find the parameters and the generator. Cohen & Welling (2014) focus on disentangling and learning the distributions of multiple compact "toroidal" one-parameter groups in the data.

**Neural Symmetry Detection** Techniques from Lie theory and generators have been used in conjunction with deep learning methods in order to identify symmetries of a task, although usually only for small angles or in a supervised setting Dehmamy et al. (2021) These are usually of interest to physicists, as this can simplify the process of identifying conservation laws or picking the right theoretical model for a given problem Krippendorf & Syvaeri (2020); Liu & Tegmark (2021). Probabilistic approaches are also of interest, especially in relation to our work, in which learning the distribution over the parameters as well as the symmetry is of interest van der Ouderaa & van der Wilk (2022); van der Ouderaa et al. (2022). Another method is found in Sanborn et al. (2022), where a group invariant function known as the bispectrum is used to learn group-equivariant and group-invariant maps from data. Benton et al. (2020) consider a task similar to ours, attempting to learn groups with respect-to-which the data is invariant, however, the objective places constraints directly on the network parameters as well as the distribution of transformation parameters with which the data is augmented.

**Latent Transformations** Learning transformations of a one-parameter subgroup in latent space (whether that subgroup be identical to the one in pixel space or not) has been accomplished by Keurti et al. (2023); Zhu et al. (2021); Roman et al. (2023). Nevertheless, other works either presuppose local structure in the data by using CNNs instead of fullly-connected networks or focus on disentangling interpretable features instead of directly learning generators that can be used as an inductive bias for a new model.

In contrast to the above, we propose a model that is able to:

- perform symmetry detection in pixel-space, without assuming any inductive biases,
- parametrize the latent space generator that stays close the one in pixel-space,
- and learn both the generator and the parameter distributions.

## 3 METHOD

The biggest technical issue to overcome in the SyMNIST task is the fact that two separate quantities need to be learned from the data: one collective, another sample-dependent. Namely, the generator and the parameter, respectively. In previous work, especially before the triumph of deep learning techniques, methods such as gradient descent (Rao & Ruderman, 1998) and expectation-maximization (Sohl-Dickstein et al., 2010) were used. In this work, we are interested in neural symmetry detection, leveraging neural networks as function approximators as was done in Dehmamy et al. (2021) and Gabel et al. (2023). We use the latent model described in the latter as a starting point.

### 3.1 PARAMETRIZING THE GENERATOR

Taking the approach used in Gabel et al. (2023) in order to parametrize the generator and allow for a broad range of symmetry transformations, we can write the generator with any specified basis for the functional form of its components. In other words, regression is performed on the coefficients of the terms of the generator. We pick an affine basis, such that the functions $\Gamma_i$ from equation 2 have the following form:

$$\Gamma_x = \alpha_{xc} + \alpha_{xx}\,x + \alpha_{xy}\,y, \quad \Gamma_y = \alpha_{yc} + \alpha_{yx}\,x + \alpha_{yy}\,y. \tag{5}$$

For clarity, in the above we replaced the integers in the subscripts by $c$ (for constant terms), $x$, and $y$ accordingly. The above affine basis can capture the "canonical" symmetries such as translation, rotation, and scaling, but many others as well. Note that one can pick an arbitrarily complicated basis for the expressions given above. This is the major appeal of this approach.

**Examples** In order to encode the three canonical symmetries, one should use coefficient matrices that yield $G_T = \partial_x$, $G_R = x\partial_y - y\partial_x$, and $G_S = x\partial_x + y\partial_y$, which are the generators of

translation (in the $x$-direction), counterclockwise rotation about the origin, and isotropic scaling w.r.t. the origin, respectively. We also choose to write the partial derivatives in the intuitive, compact notation, e.g., $\partial_x \equiv \frac{\partial}{\partial x}$. Additionally, one can write down generators for shearing, anisotropic scaling, or combinations of translation and any another of the above transformations.

To apply the above operators to a grid, one must write the partial derivatives as matrices. We use the Shannon-Whittaker interpolation, as is done in Rao & Ruderman (1998). This automatically assumes the function to be interpolated is periodic, and other interpolation schemes could have been chosen. For a discrete set of $n$ points on the real line and $I(i+n) = I(i)$ for all samples $i$ from 1 to $n$, the Shannon-Whittaker interpolation reconstructs the signal $I$ for all $r \in \mathbb{R}$ as

$$
\begin{aligned}
I(r) &= \sum_{i=0}^{n-1} I(i)Q(r-i), \\
Q(r) &= \frac{1}{n}\left[1 + 2\sum_{p=1}^{n/2-1}\cos\left(\frac{2\pi pr}{n}\right)\right].
\end{aligned}
\tag{6}
$$

To obtain numerical expressions for $\partial_x$, $Q$ can be differentiated with respect to its input. This then describes continuous changes in the spatial coordinate at all $n$ points. The above can be extended to two dimensions by performing the Kronecker product of the result obtained for one dimension with itself, mirroring the flattening operation applied to the input images. The parametrized generator hence looks like:

$$
\mathbf{G}_\alpha = \sum_{i=1}^{6} \alpha_i \mathbf{D}_i,
\tag{7}
$$

where the $\mathbf{D}_i$ are the matrices that represent $\partial_x, x\partial_x, y\partial_x, \partial_y, x\partial_y$, and $y\partial_y$.

## 3.2 LATENT MODEL

The model is a neural version of the EM-algorithm in Sohl-Dickstein et al. (2010), combined with a latent space bottleneck in order to avoid performing calculations in raw pixel-space (See Figure 2). The bottleneck has two motivations. First, learning symmetries from scratch requires learning the relationship between pixels, if one exists, in the data, and this does not scale well with input size as the space of possible connectivities is factorial in nature (cf. the permutation group). Second, the encoder should learn to remove unimportant information from the data, such as background information in an image classification setting. The autoencoder design allows for the model to keep relevant information in the latent space and transform this according to some affine mapping it shares with all other input pairs.

The learnable parameters of the model can be grouped as follows:

1. The encoder $f_\phi$
2. The decoder $g_\psi$
3. The $t$-network $T_\theta$
4. The parametrized generator $\mathbf{G}_\alpha = \sum \alpha_i \mathbf{D}_i$

The loss function consists of a part that ensures both the pixel-space and latent-space vectorized data pairs transform to each other. One can write these as:

$$
\mathcal{L}_T^X(\mathbf{x}, \mathbf{y}) = ||g_\psi \circ e^{T_\theta(\mathbf{x}, \mathbf{y})\mathbf{G}_\alpha} \circ f_\phi(\mathbf{x}) - \mathbf{y}||^2, \quad \mathcal{L}_T^Z(\mathbf{x}, \mathbf{y}) = ||e^{T_\theta(\mathbf{x}, \mathbf{y})\mathbf{G}_\alpha} \circ f_\phi(\mathbf{x}) - f_\phi(\mathbf{y})||^2
\tag{8}
$$

and can be rewritten as $\mathcal{L}_T^X(\mathbf{x}_0, \mathbf{x}_t) = ||\hat{\mathbf{x}}_t - \mathbf{x}_t||^2$ and $\mathcal{L}_T^Z(\mathbf{x}_0, \mathbf{x}_t) = ||e^{t\mathbf{G}_\alpha}\mathbf{z}_0 - \mathbf{z}_t||^2$, respectively. The notation here has been changed to emphasize the fact that the input images are related by a transformation of magnitude $t$, the hatted vector signifying the output of the model as an estimate for the transformation of the original, unaltered image. There is also a reconstruction term for both inputs in the data pair, which ignores the exponential in order to train the autoencoder, i.e. $\mathcal{L}_R(\mathbf{x}) = ||g_\psi \circ f_\phi(\mathbf{x}) - \mathbf{x}||^2$. Additionally, a term that enforces the generator in the latent space to

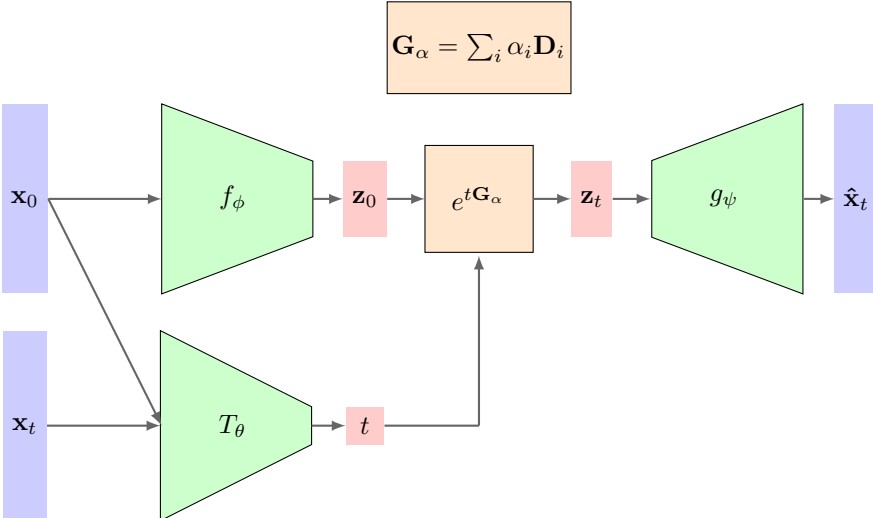

Figure 2: Latent model architecture. The inputs (blue, left) are concatenated and passed to the $t$-network $T_\theta$ that predicts the magnitude of the transformation (bottom branch). Simultaneously, the unaltered image passes through the autoencoder (top branch) and its latent vector (red) gets multiplied by the matrix exponential of the generator in latent space (orange) multiplied by the predicted magnitude, which is sample dependent. There are three neural networks (green) in the model, i.e., the encoder $f_\phi$, the decoder $g_\psi$, and the $t$-network $T_\theta$. The generator is updated through updating the coefficients $\alpha_i$, which depend on the chosen basis $\mathbf{D}_i$.

be close to the one in pixel space is introduced. This $\alpha$-*matching* term uses the learnt $\alpha_i$ from the latent space and places them in a generator for the pixel-space, using a first order Taylor expansion to compare its effect on the vectorized input, formally: $\mathcal{L}_\alpha = ||(\mathbb{I} + t\mathbf{G}'_\alpha)\mathbf{x}_0 - \mathbf{x}_t||^2$, where the prime denotes the basis $\mathbf{D}_i$ evaluated in pixel-space. The total loss, therefore, is:

$$\mathcal{L}(\mathbf{x}, \mathbf{y}) = \mathcal{L}_T^X(\mathbf{x}, \mathbf{y}) + \lambda_Z \mathcal{L}_T^Z(\mathbf{x}, \mathbf{y}) + \lambda_R(\mathcal{L}_R(\mathbf{x}) + \mathcal{L}_R(\mathbf{y})) + \lambda_\alpha \mathcal{L}_\alpha \qquad (9)$$

In order to avoid the ambiguity in the exponent of the matrix exponential ($t\mathbf{G} = st \times \mathbf{G}/s, \forall s \in \mathbb{R}_0$), the generator is normalized by enforcing the coefficient vector to have unit norm. I.e., $||\boldsymbol{\alpha}||^2 = 1$, where $\boldsymbol{\alpha}$ is the vector made up of the coefficients $\alpha_i$.

### 3.3 THE $t$, $\epsilon$, AND $z$-TESTS

Three tests were developed in order to investigate the quality of the learnt generator after training. With the $t$-test, various values for $t$ are passed to the frozen encoder-decoder part of the model, as well as a sample image. The $\epsilon$-test feeds small values of $t$ to the encoder-decoder and uses unseen test data and linear regression to calculate an estimate for the generator learnt by the model overall, this allows for a comparison to the generator learnt in latent space. The $z$-test is a visualization of the latent vector and the vector field associated with the generator learnt in the latent space.

## 4 RESULTS

A transformation and parameter distribution are chosen and fed to the SyMNIST data generator. This produces data pairs $\{\mathbf{x}_0, \mathbf{x}_t\}$ in which the first image is the original MNIST digit and the second is the transformed image. The transformation is applied using the `affine` function from the `torchvision` library. The magnitude of the transformation is the parameter sampled from the chosen distribution. This procedure allows for a lot of flexibility in testing a neural symmetry detector, as the distribution can be arbitrary and, in theory, so can the transformations. In these experiments, we will focus on detecting affine transformations.

## 4.1 PARAMETER DISTRIBUTION

The parameters predicted by the $t$-network are stored and plotted on a histogram during training (Figure 3). Beyond multimodal distributions, which the model can learn well, it seems like the distributions capture aliasing artifacts in the translation setting.

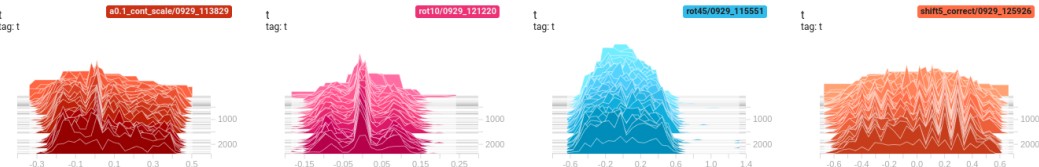

Figure 3: Parameter distributions learned by the latent model during training

## 4.2 GENERATOR PREDICTION

Investigating the quality of the learnt generator depends on what final coefficients the model converged to. We employ the $z$-test mentioned above in order to visualize the latent flow defined by the generator.

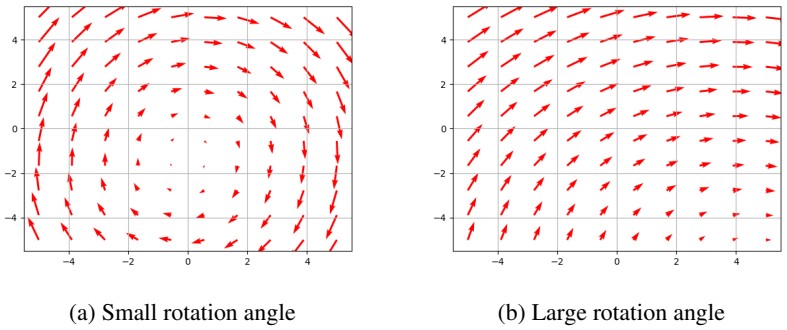

(a) Small rotation angle            (b) Large rotation angle

Figure 4: Latent flow for rotation

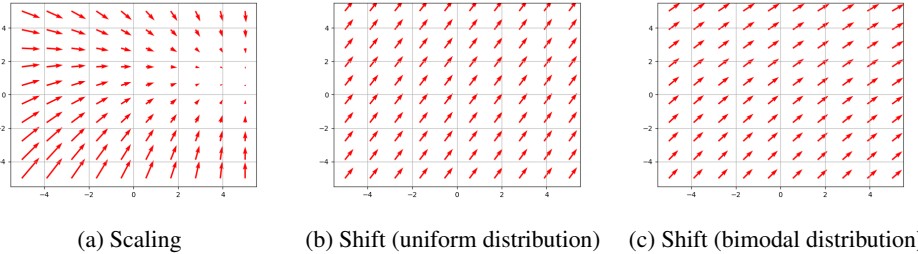

(a) Scaling            (b) Shift (uniform distribution)            (c) Shift (bimodal distribution)

Figure 5: Latent flow for other transformations

## 4.3 DISCUSSION

In most cases, it seems like the final generator has some correct values. This is probably due to the $\alpha$-matching term, which pushes the values of the coefficient towards the values expected from the first order Taylor expansion in pixel space. It is expected that this term helps the coefficients reach a basin where the correct pixel-space transformations are reachable, although placing too much weight on this term might be counter-productive for large values of the parameters.

## 5 CONCLUSION

Symmetry detection tasks rely on identifying transformations of data points that keep some task-related quality, such as classification label, identical. In this work, we proposed a latent variable framework for learning one-parameter subgroups of Lie group symmetries from observations. Our method uses a neural network to predict the one-parameter of every transformation that has been applied to datapoints, and the coefficients of a linear combination of pre-specified generators. We show that our method can learn the correct generators for a variety of transformations as well as characterize the distribution of the parameter that has been used for transforming the dataset.

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
