# OpenReview forum: "Latent Lie Group Representations"
_ICLR.cc/2024/Conference — ICLR 2024 Conference Withdrawn Submission_

### Official Review · Reviewer_8A7f · 2023-10-27

**Soundness:** 4 excellent
**Presentation:** 3 good
**Contribution:** 3 good
**Rating:** 5
**Confidence:** 4

**Summary:**

This paper introduces a method to learn symmetries in a latent representation of the data. The setting requires having pairs of inputs, one being the original and one the transformed data point. They propose a network to learn a latent space, the Lie algebra elements and the transformation parameters at the same time.
They show some limited experimental results on SyMNIST, where each image is paired with an affine transformed version of the image.

**Strengths:**

The paper discusses some good avenues for learning symmetries in data.
1. For spatial data, the vector field approach taken for the Lie algebra elements $\Gamma(x)$ (generators) is interesting.
2. The overall architecture of their model and the loss have some good points, like the $\mathcal{L}^Z$ term in eq. (8)
3. The learned generators in Figs 4 and 5 seem close to expected ground truth.
4. The theoretical sections of the paper are fairly polished, well-structured and presented well.

**Weaknesses:**

This paper is definitely moving in a good direction, but it seems to be work in progress.
1. __Lack of baselines and ground truth:__ The experiments are incomplete. They authors should compare against other symmetry discovery methods such as Augerino (Benton 2020), LieConv (Finzi 2020), L-conv (Dehmamy 2020) and recently LieGAN (Yang 2023). Similar to your eq (5), LieConv uses a subalgebra of $GL(3)$. The rotated MNIST experiment in L-Conv also dealt with pairs of input,   predicting $t$ and learning $\mathbf{G}$ in the process. LieConv and LieGAN also both use the exponential map.
2. __Novelty:__ Gabel 2023 seems to use a very similar formulation and architecture and even works with the same input data. How do your results contrast with Gabel 2023, and can their results serve as a baseline for yours?
3. __presentation of results:__
    1. Fig 3 is not clear or informative. What should the reader learn from it? What is the vertical axis?
    2. Figs 4,5, qualitative and quantitative comparison with ground truth is missing (e.g. cosine similarity with GT and plotting GT)
4. __Detials missing for reproducibility:__ I couldn't find any appendices or supplementary material. There is also not enough information in the paper (no latent dims or hyperpaparameters), which makes the paper harder to assess.

**Questions:**

1. The affine transformations you are learning, for the most part, act linearly on the inputs. Thus, I don't think going to a latent space is required in this case. Can you elaborate why you use latent space here?
2. what are the latent dims? List all hyperparameters.
3. what are the structures of neural nets $T_\theta$ and autoencoder layers?
4. Have you done ablation studies for the terms in the loss function (9)? How much do $\mathcal{L}_\alpha$ and $\mathcal{L}^Z_T$ matter?
5. $\mathcal{L}_\alpha$ needs more explanation. How does one go to the prime basis $\mathbf{D}'_i$? This may be doable if the $z$ space was also 2D, like the pixel space. But what is the exact procedure?
5. Compared to Gabel 2023, what is new in your paper?

---

### Official Review · Reviewer_bdCU · 2023-10-29

**Soundness:** 2 fair
**Presentation:** 1 poor
**Contribution:** 2 fair
**Rating:** 1
**Confidence:** 3

**Summary:**

This paper propose to learn the symmetry of a dataset via neural networks. The symmetry is represented by a one-parameter Lie group and its generator is represented and learned in the network. The sample dependent $t$ parameter is also output by the network and it is used to estimate the paramter distribution. A SyMNIST dataset is built and is used to verify the method by discovering the applied affine transformations. Qualitative results are shown.

**Strengths:**

This paper tackles an interesting topic of symmetry detection. Although it feels like the task is still at an early stage and only toy problems are tested, the related methods may become more influential after more developments.

**Weaknesses:**

1. The general writting is hard to follow for me. Many sentences are hard to decode, and mathematical notations are not explained well in texts. For example,
    - (a) The definition of "class-generic symmetries" is unclear. What exactly are "symmetries within each class, but also common among all classes"? Any concrete examples?
    - (b) The "SyMNIST task" may be better described as "SyMNIST dataset". A figure that visualizes sample images of the dataset is helpful to readers.
    - (c) "We wish to learn non-canonical symmetries". What are the non-canonical symmetries refer to? Any examples? In section 4, only affine transformations are used in experiments. Does affine transformation belong to non-canonical symmetries?
    - (d) "A collection of multiple power series applied in succession is the neural network". What is the role of non-linear activations in this description?
    - (e) "avoid making calculations in pixel-space directly". What is the calculation in pixel-space refer to? Any examples?
    - (f) "pixel-level transformation will need to be extracted by special methods". Any examples?
    - (g) The network structures of encoder $f$, decoder $g$ and t-network $T$  are not discussed.
    - (h) What is "aliasing artifacts" in the translation setting?

2. This work focuses on the one-parameter groups. What if the underlying data contains a wider range of transformations, e.g., both rotation and translation, that cannot be described by a one-parameter group?

3. There is a lot overlap with Gabel et al. (2023), including figures, equations and descriptions. It would be helpful if the authors could highlight the contribution and novelty compared to it.

4. The t-test and $\epsilon$-test are missing in experiments.

5. There is no quantitative evaluations, and there is no comparision to related algorithms.

6. Fig. 4 and Fig 5 are hard to parse and understand due to the lack of description.

**Questions:**

See weakness.

---

### Official Review · Reviewer_FZBr · 2023-11-02

**Soundness:** 3 good
**Presentation:** 3 good
**Contribution:** 3 good
**Rating:** 5
**Confidence:** 3

**Summary:**

The paper uses a symmetry identification technique to improve learning in neural networks. The mathematics underpinning the technique is robust. A good implementation would lead to improvements in neural network learning. It will also make the behaviour of NN's more transparent, as--  at least symbolically -- we get to learn what symmetries exists in data beforehand.

**Strengths:**

The use of mathematical techniques to improve data-oriented machine learning.

**Weaknesses:**

It is not clear if the implemented technique worked at all! A few figures are produced on page 7 (Figures 4 and 5), and we read that a z-test has been performed. No numerical results are however presented and I am not sure how did the model do!

**Questions:**

Does the new model learn the data better, faster, with less parameters etc etc? Can you please provide a comparison table of some sort?